# Barriers to Early Presentation amongst Rural Residents Experiencing Symptoms of Colorectal Cancer: A Qualitative Interview Study

**DOI:** 10.3390/cancers15010274

**Published:** 2022-12-31

**Authors:** Christina Dobson, Jennifer Deane, Sara Macdonald, Peter Murchie, Christina Ellwood, Lorraine Angell, Greg Rubin

**Affiliations:** 1Population Health Sciences Institute, Newcastle University, Ridley1 Building, Queen Victoria Road, Newcastle-upon-Tyne NE1 4LP, UK; 2School of Health and Wellbeing, University of Glasgow, 1 Horselethill Road, Glasgow G12 9LX, UK; 3Division of Applied Health Sciences, Section of Academic Primary Care, University of Aberdeen, Polwarth Building, Foresterhill, Aberdeen AB25 2ZD, UK; 4JJR MacLeod Centre for Diabetes, Endocrinology and Metabolism, Aberdeen Royal Infirmary, Aberdeen AB25 2ZP, UK; 5Independent Researcher, Newcastle upon Tyne NE1 7RU, UK

**Keywords:** colorectal cancer, rurality, early diagnosis, help-seeking, cancer inequalities, diagnostic delays

## Abstract

**Simple Summary:**

People living in rural areas are less likely to survive cancer than those living in urban areas. There is evidence suggesting that poorer survival amongst rural cancer patients may be because these patients take longer to be diagnosed, particularly in the time between their symptoms beginning and them going to hospital to see a specialist. This study explored the experiences of people with symptoms of bowel cancer, in rural Northern England, to examine whether there were any common causes of ‘delay’ for these patients. Participants reported that their health beliefs and self-reliance, concerns about losing time at work, and having a poor relationship with their GP, could all influence whether they consulted the GP early about their symptoms. Having identified these unique barriers for rural patients, in relation to seeing the GP about bowel symptoms, we can develop ways to support earlier consultations with GPs, to improve the outcomes of rural patients who develop cancer.

**Abstract:**

Rural cancer inequalities are evident internationally, with rural cancer patients 5% less likely to survive than their urban counterparts. There is evidence to suggest that diagnostic delays prior to entry into secondary care may be contributing to these poorer rural cancer outcomes. This study explores the symptom appraisal and help-seeking decision-making of people experiencing symptoms of colorectal cancer in rural areas of England. Patients were randomly invited from 4 rural practices, serving diverse communities. Semi-structured interviews were undertaken with 40 people who had experienced symptoms of colorectal cancer in the preceding 8 weeks. Four key themes were identified as influential in participants’ willingness and timeliness of consultation: a desire to rule out cancer (facilitator of help-seeking); stoicism and self-reliance (barrier to help-seeking); time scarcity (barrier to help-seeking); and GP/patient relationship (barrier or facilitator, depending on perceived strength of the relationship). Self-employed, and “native” rural residents most commonly reported experiencing time scarcity and poor GP/patient relationships as a barrier to (re-)consultation. Targeted, active safety-netting approaches, and increased continuity of care, may be particularly beneficial to expedite timely diagnoses and minimise cancer inequalities for rural populations.

## 1. Introduction

Rural cancer inequalities have been evidenced internationally for decades, with rural residents 5% less likely to survive cancer than their urban counterparts [1]. These inequalities are noted across a range of cancers, including colorectal cancer (CRC) [2]. CRC is the second greatest cause of cancer mortality both globally [3] and in the UK [4] and CRC patients living in rural areas experience poorer outcomes than those living in urban areas [5]. Time to diagnosis is associated with cancer survival more broadly [6,7], with delays across the diagnostic pathway impacting cancer outcomes. There is mounting evidence to suggest that rural populations experience prolonged diagnostic intervals, in comparison to their urban counterparts. Rural cancer patients have more advanced disease at diagnosis [8,9] and are less likely to be in receipt of a diagnosis at the time of death than urban patients [10], suggesting that diagnostic delays could be a cause of rural cancer inequalities.

Although screening is an effective tool for reducing CRC incidence, in most European countries less than 30% are detected through screening [11], with screen-detected CRCs accounting for fewer than 10% of cases in the UK [12]. The majority of CRCs are diagnosed after patients present with symptoms (either in primary care, or as emergency presentations), with over 55% being diagnosed after patients present to, and are referred by, a General Practitioner (GP) [12]. The cancer diagnostic pathway (in the context of symptomatic presentations) spans the period from a patient’s symptoms first beginning up to the point of diagnosis (or commencement of treatment). This pathway consists of three component stages: the patient interval (time from symptom onset to first presentation), the primary care interval (the time from first presentation to specialist referral), and the secondary care interval (time from specialist referral to diagnosis) [13]. 

In the secondary care interval rural and urban patients appear to have similar times to diagnosis [14], with some of the most remote patients experiencing shorter secondary care intervals than their urban counterparts [15]. Rural cancer patients are more likely to have 3 or more consultations prior to referral, with longer primary care intervals reported by rural GPs, than urban practitioners [16]. Rural cancer patients are also more likely to have a cancer “alarm” symptom at first presentation [16]. 

The impact of travel time, and distance, has been examined as a potential cause of rural cancer inequalities, however, the evidence is not consistent. Whilst cancer patients in rural Western Australia reported that travel time was a barrier to early presentation [17], other research has shown that travel time to primary care is not associated with tumour stage at diagnosis [16]. 

Whilst existing evidence points towards the patient interval and primary care interval as possible sites for diagnostic delays for rural patients, we do not yet know what these delays may be, or why they are occurring. Given the complexity of patient decision-making in the context of cancer symptoms, it is possible that rural patients experience barriers to early presentation and diagnosis that have not yet been described. 

RURALLY (Recognition, Understanding of, Responses to, bowel symptoms Among people Living in rural Localities of Yorkshire) examined the experiences of rural residents experiencing symptoms of possible CRC, aiming to understand help-seeking decision-making processes, and identify potential barriers to early diagnosis, and sources of diagnostic delays, for this population. This mixed- methods study utilised postal surveys and semi-structured interviews to examine symptom appraisal and help-seeking decision-making amongst rural dwellers who experience symptoms of possible CRC. Here, we report the findings of the interview phase of this study. 

## 2. Materials and Methods

### 2.1. Study Design

This paper reports findings from the interview phase of RURALLY, a mixed-methods study. Within this study, we utilised semi-structured interviews to examine symptom appraisal and help-seeking decision-making amongst people residing in rural areas, who had recently experienced symptoms of possible CRC. 

### 2.2. Population and Recruitment

Four practices in North Yorkshire, England, serving rural communities of differing sizes and characteristics, were selected as sites for recruitment. Although much research has treated the ‘rural’ as a homogenous population, simply one half of a rural/urban dichotomy, rural residents and communities are in actuality very diverse in relation to a range of characteristics, including size, age, gentrification, in/out migration, affluence/poverty and local economies [18]. The Office for National Statistics (ONS) categorises rural areas as belonging to one of 6 categories, with rural town and fringe being the least remote and hamlets and isolated dwellings the most [19]. We selected primary care practices that served communities of differing degrees of rurality and remoteness to enable us to explore the breadth and depth of rural experiences, the characteristics of which are outlined in Table 1. 

Primary care staff at each site generated a random list of patients aged 40 and over and sent a study pack (containing participant information sheet, consent form and survey) to the first 850 eligible patients. The survey asked participants whether they had experienced any bowel symptoms in the preceding 8 weeks, their initial appraisal of these symptoms (i.e., thoughts on severity and impact), and whether or not they had consulted a health professional about their symptoms. 

Recruitment commenced in November 2019, with practices 1–3 inviting patients consecutively and fieldwork being undertaken until March 2020. Due to the pandemic, research activities were paused and patients from practice 4 were invited to participate in January 2021 and interviews completed by the end of March 2021. A total of 720 surveys were returned (21% response rate). 

A sub-sample of respondents were invited to take part in an interview. The survey responses facilitated purposive sampling of interviewees [20], enabling us to engage with a range of symptomatic experiences, as well as seeking diversity in social and geographical characteristics. As fieldwork progressed a number of participants were theoretically sampled [21] for occupation, as self-employment (particularly in the farming and tourism industries) emerged as an area which influenced the patient interval. 

### 2.3. Data Collection

Interviews were semi-structured. A semi-structured approach provided consistency, ensuring that key topics were covered across all interviews, given that this is an under-researched area, whilst also allowing the researcher the ability to probe further into certain topic areas and the freedom to explore novel topics of relevance that arose during the research encounter [22]. The topic guide was collaboratively developed by the authors, who have collective expertise in cancer, primary care, sociology, psychology and qualitative research. Our study lay representative (LA) also contributed to the development of the topic guide, and piloting of the questions. The topic guide was iterative, and as such was revisited and refined after the first two interviews, as well as later in the fieldwork period, as novel topics of relevance arose during interviews. 

The topic guide provided a structure of key topics which we wished to cover in interviews, without being prescriptive about the exact phrasing of interview questions. This allowed the researcher to tailor questions to specific research encounters, for instance relating a question to the wider narrative being shared. Participants were asked about the symptoms they experienced, the impact of these on their daily lives, thoughts about possible causation and whether or not they consulted about their symptoms (and, if so, an estimation of patient interval length was obtained) (see Appendix A to view the topic guide). Interviews also explored beliefs about accessing health care services, and how rurality influences this, as well as experiences of health and illness in a rural setting more generally. It is important to note that these participants had only reported experiencing symptoms of CRC, meaning the majority did not have a CRC diagnosis, and a number had not consulted about their symptoms. Therefore, during the interviews the word “cancer” was not used by the interviewer, and it was only discussed if brought up by the interviewee. 

Participants from practice 4 were interviewed after the onset of COVID-19 and, as such, additional questions were included in the topic guide, to explore how remote consultation affected help-seeking decision-making and willingness to engage with health care services. 

A total of 40 interviews were undertaken. Interviews with participants registered at practices 1–3 (*n* = 26) were face-to-face, and predominantly took place in participants’ homes, though 3 interviews took place in coffee shops, at the interviewee’s request. Recruitment and fieldwork at practice 4 were delayed by COVID-19 restrictions until early 2021 and all interviews at the final practice (*n* = 14) were conducted remotely. Participants were offered the choice of a telephone interview or interview via a video conferencing platform (i.e., Zoom or Microsoft Teams). All 14 participants opted for a telephone interview. 

Interviews were conducted by one of three interviewers (CD, JD or CE), were audio recorded (on an encrypted digital dictaphone) and lasted an average of 49 minutes (range of 28–76 min). Interviewing ceased when new interviews repeatedly supported the emerging theory, did not challenge the analysis, and a point of ‘accuracy’ was felt to have been reached within the dataset [23,24].

### 2.4. Data Analysis

Audio recordings of the interviews were pseudo-anonymised and transcribed verbatim by a professional transcription service. Transcripts were checked for accuracy against audio recordings and any potentially identifiable information was redacted or pseudo-anonymised. 

Analysis occurred concurrently with fieldwork, to support sampling and probing of later interviewees. The first 3 transcripts were read and re-read, then coded, by 2 researchers (CD and CE). Initial codes were compared, discussed and refined to generate a single coding matrix that was used with subsequent transcripts. Like the topic guide, the coding matrix was iterative, and as new codes were identified, they were incorporated into the matrix and previous transcripts revisited to identify their presence. Transcripts were coded and organised in NVivo software.

A number of strategies were employed to embed rigor within the analysis process, including: constant comparison within and between transcripts, and codes; memo writing; discussion of deviant cases; and regular interviewer meetings to discuss emerging data. Transcripts, codes, themes and sub-themes were explored and developed in depth during regular meetings of the RURALLY qualitative sub-group (CD, JD, CE, LA & SM). During these meetings we collectively looked at the qualitative data, drawing on the different experiences and perspectives of team members to achieve an interpretation of the data that was felt to be representative of participants’ narratives and the wider dataset. 

During the analysis phase we sought to be reflexive about how our characteristics shaped the interview encounter (i.e., ‘outsider’ status [25]) and the ‘sensitizing concepts’ [21] we brought to the analysis, because of our *a priori* knowledge of rural cancer inequalities. The researchers being ‘outsiders’ to the community was felt to be beneficial in research encounters, as it enabled us to ask participants to explain their communities and concepts of rural living in greater depth than they may have done with someone who they saw as an ‘insider’ with whom there would be shared knowledge and points of reference. Much time was spent developing a coding matrix which was thought to be representative of the data, whilst also mindful of the sensitizing concepts brought to the research. The iterative nature of the matrix supported an inductive approach to data analysis, whereby findings were rooted in the narratives of participants, and not forced into a pre-conceived framework. 

Ethical approval for this study was obtained from the NHS National Research Ethics Service (NRES) Wales REC 1 Committee (REC Ref: 19/WA/0198). 

## 3. Results

Forty participants, with varied demographic characteristics took part in an interview (see Appendix A). 

Analysis of the data identified several themes which were key in participants’ narratives of both willingness and timeliness of presentation. These were: a desire to rule-out cancer; stoicism and self-reliance; time; and GP/patient relationships. 

### 3.1. Desire to Rule out Cancer

Participants who were concerned that their symptoms could be caused by cancer were eager to consult early, particularly when symptoms were coupled with a family history of bowel cancer.


*“My father had cancer and died early. He was 54 when he died. He had stomach cancer. And it, basically, spread and his entire body shut down. So, I’m starting to be more wary about that around the genetics and all the rest of it. And because I had the stomach pains as well. And because on my father’s side that history made me just a little bit… but it was mainly because I saw the blood in the stool that I thought this isn’t something just normal. And because of the length of time that I was having the trouble as well that I thought, yeah, I need to go to the doctor and check it all out and stuff like that.”*

*P11*



*“My dad had bowel cancer and died of bowel cancer and my sister had had a spell of bowel cancer and she is ten years older than me, and she has had part of her bowel removed so I wanted, because of my anxiety and stressing out about it, to go get it checked out, so I was straight down to the doctors.”*

*P18*


### 3.2. Stoicism and Self-Reliance

Life in a rural environment can be taxing and requires people to be stoic and self-reliant in an array of contexts, which also translates into their approach to the management of illness. Participants discussed how people in rural areas generally take a pragmatic approach to managing episodes of illness, and that going to the doctor for minor ailments could be seen as weakness.


*“I think you just take this, kind of, no nonsense approach to everything, including bad health. There is an element of “oh just take yourself in hand and give yourself a stern talking to and you’ll be fine”.”*

*P38*



*“We have to face challenges as they come along. We can’t rely on the city council…We are on our own out here. If our road floods, we have to dig the ditch to let it drain off down the river. So, we are probably quite self-sufficient. And I was ill for about five weeks in January of this year with proper flu…I didn’t bother going to the doctor or anything. I mean, I know how to look after myself.”*

*P10*



*“I mean, you don’t hear of people in the Dales going to the doctor’s, just, they won’t. Yes, and I think not to be healthy is seen as a weakness.”*

*P14*


There was a strong sense that living in a rural area heightened people’s connection with nature, particularly for those involved in farming, which had a positive influence on people’s knowledge of, and ability to manage, illness. 


*“Farmers understand animals and that passes on to any sort of illness, you know, and you tend to be probably a bit more resourceful.”*

*P09*


Managing oneself during periods of illness was seen as characteristic of rural populations and whilst some participants felt that these outlooks were most evident amongst “native” rural residents, others felt that they developed over time, as a consequence of adjusting to, and navigating, life in a rural environment. 


*“Old-fashioned country people are certainly more…stoic about their illnesses, so maybe they hang on a bit longer before they go and see their GP.”*

*P02*



*“I think they [“native” rural residents] tend to be more self-reliant and stoical, more willing to accept the ups and downs of their health and declining abilities.”*

*P08*



*“You’re more in touch with nature and I think that’s beneficial for your health, but you start to notice the changes in the atmosphere and you start to be able to tell when the rains going to come or it’s going to snow. You trust your instincts a bit more. You become less reliant on other people’s opinions and other people’s outlooks I suppose. You become more self-sufficient.”*

*P05*


### 3.3. Time

Time was seen as a valuable commodity for both rural residents and health care practitioners and could be a barrier to consultation for people experiencing symptoms of CRC. 

Long travel times to access primary care were common for participants and this served as a barrier to presentation for some, particularly those who were self-employed. The length of travel time itself was a secondary problem to the subsequent time lost in the workplace, resulting from the combined time required for the consultation and return journey. This was particularly true for people who worked in seasonal industries, such as tourism and farming. 


*“A lot of people round here are self-employed, and you don’t have that…it’s not quite so easy is it? If you’re employed and you need to go to the doctor, you can just go to the doctor. You’re entitled to go to the doctor and turn up late for work….You’ve got every right to go to the doctor haven’t you without losing any pay or anything.”*

*P26*



*“There are farmers who think ‘Well, I just can’t get there because it’s lambing time or it’s hay time, or I’ve got to get the stock fed’.*

*P40*


Not wanting to waste time was a barrier to consultation that was discussed by a number of participants. People were reluctant to waste the GP’s time by consulting about conditions that could be self-managed, or which the GP may think of as minor. 


*“I don’t want to waste his time when I don’t need to. Because I consider myself as being the first line of defence against if I’ve got any problems. It’s up to me to start off and look and see if I can find anything about it and what can I do about it before I go to see you (the GP).”*

*P34*



*“I can’t even go and say “I’ve got an excruciating pain”, it’s just a dull…it’s very uncomfortable sometimes but it’s only for an evening….So it just feels like I’m going and wasting his time really…he might be wondering “What is she here for?” because this is quite minor.”*

*P26*


Participants also talked about not wanting to waste their own time by consulting about symptoms that they felt the doctor would be unable, or unwilling, to do anything to resolve. This was particularly the case when people had previously consulted about the symptom and believed that re-consulting would be ‘a waste of time’. 


*“I haven’t been for a while about it because I think it was a waste of time. I think last time I went she just said, “Oh well, just carry on what you’re doing, you’re doing okay and there’s nothing to worry about.” Because I noticed I was bleeding and I said, “Oh God, what’s going on here?” And then she just said, she gave me some cream and said, “It’s probably just piles or whatever, and that’ll settle down.” So I’ve never really gone back because I thought, “Well, I didn’t get any answers that I really wanted”.”*

*P04*


### 3.4. GP/Patient Relationships

A common theme in participants’ accounts was how the relationship between the individual and the GP served as a mediating factor in their willingness to consult about symptoms. When people felt that they had a good relationship with their GP, characterised by feeling like they knew their GP, and that their GP knew them and took a holistic approach to their care, they were happy to consult. 


*“Everybody in the village knows that he runs over time wise, but you come out feeling he’s actually listened.”*

*P06*



*“He [the GP] knows you, he’s probably read up before you walk into the surgery. Probably looked down through your notes on his computer but there’s certainly a knowledge around you and you can chat and talk to him quite happily. You never feel rushed, you maybe get seven, eight, ten minutes, something like that, but you never feel rushed.”*

*P34*


Conversely, when people felt that they did not have a good relationship with their GP they were reluctant to consult, or re-consult, about symptoms. Participants reported feeling like they were rushed through consultations and they contrasted their current experiences with stronger relationships they had had with historical GPs, who they felt had a greater understanding of their patients’ broader medical and social histories. 


*“We used to have, our GPs had been there a fair while and get to know you, so when you go in they know that you don’t go unless you’ve got to. They knew the history of the patients…Now every time you go it’s someone different…I just have no faith in them at all.”*

*P15*



*“I know the GPs don’t have much time once you enter the room really. Yes, you just feel as if [you would rather be seen by] somebody who knows like a more holistic approach, rather than turning up with some sort of symptom and them discussing that with you.”*

*P16*


Perceptions of the strength of relationship with the GP, and subsequent impact on willingness to consult, differed in relation to the size of the practice. Patients registered at practices with smaller list sizes, in smaller rural communities, generally reported strong relationships with their GP, whereas patients registered at larger rural practices were more reluctant to consult about symptoms. 


*“Because it’s not a large surgery you… it’s just a smaller surgery, so when you get smaller things, you’re able to spend more time or seem a little bit more family type of thing…I think it purely is because it’s a rural setting. It’s a smaller surgery, so everybody knows each other. You weren’t a face when you go into reception.”*

*P11*



*“The main principal guy who was my doctor…talk about trying to push a wheelbarrow up a hill in deep sand to actually get the job done, and cavalier and dismissive and really unhelpful. I got to the stage where I said to my wife “I am not going to see that bloke ever again unless I’m hopping in on one leg, carrying my other in my arm”.”*

*P07*


Whilst problematic GP/patient relationships were most commonly reported by participants registered at larger rural practices, what was notable was that these reports came from “native” rural residents, who mourned the loss of the ‘old fashioned-touch’ of their historical rural GP, whereas patients registered at the same practices who had migrated into the area did not report poor relationships with their GP, nor did they cite their relationship with their GP as a barrier to consultation. 

## 4. Discussion

Analysis of interview data from the RURALLY study identified four key themes that influenced help-seeking decision-making and willingness to consult. A desire to rule-out cancer as a possible cause of symptoms served as a facilitator to early presentation, whereas stoicism and self-reliance, as well as time scarcity and concerns about time wasting, were barriers to presentation. Participants’ perceptions of their relationship with the GP could serve to facilitate, or hinder, presentation, with ‘poorer’ relationships being most commonly reported in larger rural practices. 

For participants in this study, concerns that cancer may be the cause of their symptoms was a key motivation for consultation, and other research has also shown that anxiety, or fear, about an underlying cancer acts as a prompt to consultation for many [26,27]. 

Self-management of symptoms, and stoicism about episodes of illness, was a core element of accounts from rural residents in this study. A tendency amongst rural participants to manage episodes of illness independently and only consult once symptoms became unmanageable, may explain why rural patients are more likely to present with cancer alarm symptoms or through emergency routes [5]. Work in rural Western Australia has found that stoicism and machismo are core to many rural peoples’ identities, and a desire to present as being ‘tough’ can result in a reluctance to consult about symptoms, so as not to be seen as weak [17]. However, these beliefs have not been found in all rural Australian communities [28].

Self-reliant beliefs and behaviours have been identified more broadly within rural populations, with rural masculinity constructed around the image of the ‘strong farmer’ stereotype [29]. Moore et al. (2012) discuss how rural populations are commonly identified as ones in which there are *‘geographies of stoicism’,* and how stoicism is a sociocultural phenomenon constructed through a combination of factors, including environment, community and occupation [30]. When applying stoicism as an explanation for prolonged patient intervals and later presentation, we should be mindful that this construct does not exist independently of social structures, institutions, policies and identities, just in the same way that we should avoid depicting rural populations as homogenous, and perpetuating and privileging the stereotype of the rural, white, male farmer [31]. In this research we strove to engage with the accounts of a range of individuals, reflective of the heterogeneity of rural communities and residents to develop a rich and detailed understanding of ‘the rural’ as a complex and nuanced population. 

‘Time’ has long been hypothesised to be a barrier to early presentation for rural cancer patients, with long travel times to both primary and secondary care common for the most rural and remote patients [5,32]. Long travel times have been found to be a barrier to help-seeking among symptomatic patients, with individuals often waiting to consult about symptoms until they can combine the journey with other activities that need to be done in that area [17]. Amongst individuals experiencing acute chest pain in Australia, the most rural and remote patients were found to wait the longest before accessing emergency care [33]. 

We found that the time burden of the actual journey was less consequential in participants’ help-seeking decision-making, than the implications of the overall time lost as a result of consulting. We found that participants working in tourism and farming found the time costs of consultation to be a notable barrier to help-seeking and this was particularly acute during periods of seasonal pressure, such as in peak-tourist season, or during lambing or harvest periods, when time away from the workplace had significant financial and practical consequences. In rural areas there are often high levels of self-employment [34] and self-employment has previously been found to be associated with prolonged patient intervals for rural populations [17].

The theme of ‘time wasting’ as a barrier to help-seeking was a consistent thread across participants’ accounts. Not wanting to waste the doctor’s time for symptoms, or conditions, which may be perceived of as trivial has been reported extensively in the literature to date [35,36] and is rooted in a desire to not appear ‘foolish’ to the GP, as well as an awareness of an already over-stretched, and finite, NHS resource [37]. A novel finding from this study was that participants also identified consultation as an example of time-wasting in relation to their own personal time. Consultation was felt to be a waste of time, either because of competing priorities, such as work, or because they believed that the GP would not do anything to alleviate, or manage, their symptoms, even if they did consult. 

Participants’ perception of their relationship with their GP was a key theme in the willingness of many to consult about their symptoms. ‘Good’ relationships were characterised by a feeling that the GP knew the individual and their medical history and took time over their appraisal in the clinical encounter. These positive relationships were consistently reported at smaller rural practices, with participants feeling that these relationships encouraged them, and made them feel confident, to consult about a range of symptoms. Poor relationships with GPs were generally reported at larger rural practices, and most commonly amongst “native” rural residents, as opposed to participants who had migrated into the area, often during retirement. Participants reporting poor relationships with their GP often felt that they were not viewed or treated holistically, or given any real time in the clinical encounter, and contrasted these experiences with the care they received from previous GPs, who they felt knew their medical and social history, and provided a more personalised care service. 

The relationship between a patient and their GP has been noted as influential in help-seeking behaviour, with patients preferring to see a doctor who knows their personal and medical history [38] and preferring to wait to consult their named GP, even if this means waiting longer to be seen [39,40]. Participants in this study mourned their ‘closer’ relationships with previous GPs, whom they felt appraised, managed, and referred on for, conditions more skillfully and with a higher degree of concern for the patient’s best interests. Personalised relationships with GPs may be advantageous from a clinical perspective too, as the length of a relationship between a GP and patient is significantly associated with use of out of hours services, acute hospital admissions and mortality [41].

Participants’ perceptions of their relationship with their GP also influenced how willing they were to re-consult about symptoms that had persisted, changed, or worsened. In these instances, many felt that re-consulting was pointless, as the GP was unlikely to do anything which would help. Other research has found that rural CRC patients report primary care delays as a result of the GP dismissing their symptoms as unremarkable [28]. A lack of continuity of care seemingly makes it more difficult for patients to re-consult [42] and was also found to be associated with time to diagnosis for CRC patients [43]. 

Given the reluctance to re-consult, and a broader stoicism about illness, found amongst participants in this study, the active management of patients presenting with potential symptoms of CRC in rural areas requires additional attention. Research undertaken with GPs has shown that responsibility for the arrangement of follow-up appointments, or re-consultation, is often seen to lie with the patient [44], however, our data suggest that adoption of more active safety-netting approaches may be beneficial for rural populations. Active safety-netting approaches, including scheduling of follow-up appointments, communication of uncertainty, along with advice about how and where to seek-help, are thought to be particularly beneficial amongst patient groups who are less likely to re-consult [45] and patients have been found to prefer active safety-netting approaches [46]. 

Whilst this study focused on CRC symptoms the findings may be applicable to other cancer sites and could more generally inform strategies to support rural patients in engaging with health care services. Future research should work with rural communities to develop such strategies, ensuring that the heterogeneity of rural communities, services and environments is reflected, and the community is at the heart of efforts to reduce cancer inequalities and improve survival. 

## 5. Conclusions

This study identified 4 key themes that influenced help-seeking decision-making and consultation for rural residents experiencing symptoms of CRC. A desire to rule out cancer was a prompt to presentation, whereas scarcity of time and stoicism were barriers to presentation. The relationship between patients and their GP could either facilitate or hinder earlier presentation, with ‘good’ relationships between patients and GPs making barriers to consultation easier to overcome. Poor relationships with GPs were reported as a barrier to presentation by patients registered at larger practices, and most commonly by “native” rural residents. Continuity of care may be particularly important for these patients to facilitate early diagnosis and an active approach to safety-netting may encourage these patients to feel valued and supported to (re-)engage with primary care. 

The findings of this study highlight the barriers to early presentation that people in rural areas, who are experiencing symptoms of CRC, face, however, these findings are also potentially applicable to other cancer sites. Interventions to support earlier diagnosis of cancer amongst rural patients could be informed by the barriers identified in this study, whilst recognizing the different behavior and symptomology of cancer at different sites.

## Figures and Tables

**Table 1 cancers-15-00274-t001:** Primary Care Recruitment Site Characteristics.

	Practice 1	Practice 2	Practice 3	Practice 4
Rural Category of the GP Pratice	Rural Village & Dispersed	Rural Town & Fringe	Rural Town & Fringe in a Sparse Setting	Rural Village & Dispersed in a Sparse Setting
Practice List Size (at time of fieldwork)	8262	7303	7976	1587

## Data Availability

As per ethical approvals for this study, participants were not asked to provide consent for their interview data to be shared beyond the research team, due to the personal and potentially sensitive nature of the interview topics. As such, the interview data set is not freely available.

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
