# Peer review of "Barriers to Early Presentation amongst Rural Residents Experiencing Symptoms of Colorectal Cancer: A Qualitative Interview Study"

_cancers, 2022, doi:10.3390/cancers15010274_

Round 1
Reviewer 1 Report (Previous Reviewer 2)
Thank you for having clarified your procedures, which helped clarify your study. However, the framework is strictly local and the methodology followed aimed only at signaling possible hypotheses (although if experienced in practice) whose burden ought to be verified by a sound epidemiological "deductive" study.
The study does not provide a comparison between rural and urban areas: It could have been helpful to understand what is actually related to the "rural" residence or to other causes common with the urban one. Anyway, it is likely that these differences might not be as pronounced in this context than what is happening in larger and poorer Countries.
Divergent views on the discussion are still on the table:
Although it is essential to break down obstacles to a prompt medical consultation (and taking charge by public health services) at the onset of symptoms, only an active screening program can effectively decrease colorectal cancer incidence (and not only mortality, due only to the improvement of therapy). This should be highlighted in a high-income country.
Colorectal cancer symptoms are very not specific and, if a cancer is present, late (for example: macroscopic bleeding following ulceration of the lesion, constipation and diarrhea following stenosis). The cited study of Loo et Al examines “symptoms” in a very general approach: we cannot consider stage IV cancer trend as an index of cancer prevention of colorectal cancer in a high- income country as England is.
Author Response
We agree with Reviewer 2 that only screening will reduce CRC incidence (through detection of precursor adenomata) and that it is valuable for reducing mortality by identifying early stage, asymptomatic disease. However, in most countries in Europe, less than 30% of cases of CRC are detected though screening (Cardoso et al, Lancet Gastroenterology and Hepatology 2022) and early assessment of the symptomatic patient remains important in reducing mortality. We have added a sentence into the introduction to acknowledge the value of screening in reducing incidence, whilst situating this research in the broader context of symptomatic presentation/diagnosis that prevails across Europe.
We disagree with reviewer 2’s rather fatalistic view that symptoms indicate late disease. Koo et al (Lancet Oncology 2020) have shown that for the commonest presenting symptoms of CRC (rectal bleeding, alteration in bowel habit, abdominal pain), over 70% of cases have stage 1-3 disease and are therefore candidates for curative surgery.
Reviewer 2 Report (Previous Reviewer 1)
Dear Authors,
Good job with the revision of the paper.
Some few comments;
Page 4, line 137:
"Out study lay representative also contributed...." do you mean Our study?
Its the "topic guide" similair as the interview guide? Why a topic guide instead of a interview guide? please clarify
Table 2: Its participants ID the same as the ID in the trial? My recommendation is to change the "ID" to "clean" numbers instead.
Author Response
|
Reviewer 2 |
|
|
"Out study lay representative also contributed...." do you mean Our study? |
Thank you for identifying this typo, we have now rectified this in the manuscript |
|
Its the "topic guide" similar as the interview guide? Why a topic guide instead of a interview guide? please clarify |
We decided to use a topic guide for this study as it provided a structure of topics to be covered in each interview, whilst also allowing the researchers to have the flexibility to re-phrase questions in a manner most appropriate for each interview encounter, for instance to be able to refer it back to something earlier discussed etc. The interview topic guide allowed us to ensure that all questions were explored with participants but the exact wording of these could be phrased to most suit the specific encounter. We have now added in further information justifying the choice of a topic guide over a more structured interview guide.
|
|
Table 2: Its participants ID the same as the ID in the trial? My recommendation is to change the "ID" to "clean" numbers instead.
|
We take on board reviewer 2’s comments and have amended the participant ID numbers as suggested – a revised participant characteristics table has been uploaded and references to participant numbers in the findings amended accordingly. |
This manuscript is a resubmission of an earlier submission. The following is a list of the peer review reports and author responses from that submission.
Round 1
Reviewer 1 Report
Dear Author,
Thank you for the opportunity to read your study.
Some questions and reflections related to your paper.
The overall challenges and questions that I have as a reader;
-Title: Maybe add information about the method/design.
-Introduction: Not a clear knowledge gap and rationale, which emphasizes the importance of this qualitative study. I also recommend clarifying the aim and specific research questions related to the aim. Or is this a mixed method study? Or is the quantitative part reported elsewhere?
-Methods: Please clarify the design, now it is more an aim under the heading Study Design.
Vague qualitative method description, maybe related to the quantitative part included.
Please include the interview guide, and the specific questions you used.
Why you choose grounded theory? and use a semi-structured interview? (often open ended interviews), and then mention the “freedom to probe”?
Do you pilot test the interview guide?
-Did you reach data saturation?
-Can you please explain rigor and the reflexivity?
- How much data? Time durations of the interviews?
Please add example of the data analysis process (coding matrix), and how your work in the team with your transcripts and analysis.
An interview and transcripts can never be anonymous, please revise to pseudo anonymous (maybe you coded the transcripts?)
- The Table 2 it is hard to read. It is possible to clarify? Add medical/clinical information? What is IMD Decile? Add social status?
To improve your demographic description of the participants in the study, improve the quality of credibility of the study/transferability.
Maybe present participant 1, 2, 3 and all the related information in a table- to make it easier to relate your quotations to specific informants.
-Do you have any themes? “Factors” seemed to be more of a quantitative terminology? The result section is sliced, with “several factors”. Overall, a lot and rewarding quotes, the paper would benefit from having in-depth analysis text. As reader it seemed to be sliced, especially with 40 interviews, but not clear how much data do you have.
Discussion: I miss a deeper discussion about rigor and trustworthiness with adequate literature, please add.
The conclusion it is very long and more than a conclusion, my recommendations is to synthetize and highlight the specific findings and clinical implications from this specific study.
Reviewer 2 Report
The study highlights the different cancer symptom awareness and the propensity to consult the GP in different groups of rural population. The impact of these factors on cancer prognosis problem is well known from the literature: in general, the socio-economic status drops, the worse the outcomes. The study was designed to investigate this issue in a specific area of interest, but only a brief descriptive analysis was performed (one table) and results are presented by narrative fragments of 40 interviewee answers (out of 720 responders from 3,400 target patients) without any structured analysis.
Here are some further weaknesses:
a) Nothing is said about the 720 responders to the questionnaire: were there any symptoms to choose from the questionnaire or did the patients have to describe them (with what criteria of relevance to the CRC)? What about answers for each rural category? What about possible selection biases of responders versus not responders? Why was this first phase of the work not analyzed?
b) Do you know which of the interviewees had a cancer diagnosis between the compilation of the questionnaire and the interview (and at what stage)? How was a possible recall bias managed between those who have in the meantime been diagnosed with cancer and who have not?
c) A comparison with an urban resident cohort could have been interesting.
Main important issues that need to be considered in introduction/discussion paragraphs:
a) The paper focused the need to improve awareness to cancer symptoms “to improve the outcome of rural patients” through the experiences of citizens with “symptoms” (which?) of CRC. As said, this rule is obviously “applicable to other cancer site”, but with very different efficacy, starting from CRC, whose carcinogenesis, in most cases, offer the best chance to avoid the onset of invasive cancer by treating premalignant lesions, besides “to improve the outcomes of rural patients who develop cancer”. In cases of aggressive cancers (as CRC) symptoms are often late and they may not significantly improve patients’ outcome
b) Focusing the CRC diagnostic path, why nothing is said about screening programs (the word “screening” was never mentioned in the paper”)? European Guidelines recommend screening (in age 50-74 for average risk people and from 40 yrs for high risk one) as the best strategy to prevent cancer onset and death. Screening also has problems of adhesion among deprived citizens, but sensitizing them to screening is more appropriate than asking them to interpret symptoms that are often unspecific.
c) For a substantial proportion of all incident cancers as CRC, cervix uteri and female breast cancer, screening represents also a model of a public health that actively offers effective and appropriated taking charge of citizens’ needs, instead of waiting them at the door of a clinic. Interviewees answers told also of time wasted for inappropriate approach or problems in continuity and proximity of services: these are problems with public health organization and not of citizens’ awareness.
d) In the “results” section, the periods (plain text) between the interviewee answers' responses (in italics) seem to belong more to the comments than to results presentation.
Two minor problems:
a) In table 2 the meaning of “IMD decile” is not clear.
b) Please check for typos in the text (e.g. rows 26, 115, 134, 163)